# Misic, a general deep learning-based method for the high-throughput cell segmentation of complex bacterial communities

**Swapnesh Panigrahi[1], Dorothée Murat[1], Antoine Le Gall[2], Eugénie Martineau[1], Kelly Goldlust[1], Jean-Bernard Fiche[2], Sara Rombouts[2], Marcelo Nöllmann[2], Leon Espinosa[1], Tâm Mignot[1]\***

[1]CNRS-Aix-Marseille University, Laboratoire de Chimie Bactérienne, Institut de Microbiologie de la Méditerranée and Turing Center for Living Systems, Marseille, France; [2]Centre de Biochimie Structurale, CNRS UMR 5048, INSERM U1054, Université de Montpellie, Marseille, France

**Abstract** Studies of bacterial communities, biofilms and microbiomes, are multiplying due to their impact on health and ecology. Live imaging of microbial communities requires new tools for the robust identification of bacterial cells in dense and often inter-species populations, sometimes over very large scales. Here, we developed MiSiC, a general deep-learning-based 2D segmentation method that automatically segments single bacteria in complex images of interacting bacterial communities with very little parameter adjustment, independent of the microscopy settings and imaging modality. Using a bacterial predator-prey interaction model, we demonstrate that MiSiC enables the analysis of interspecies interactions, resolving processes at subcellular scales and discriminating between species in millimeter size datasets. The simple implementation of MiSiC and the relatively low need in computing power make its use broadly accessible to fields interested in bacterial interactions and cell biology.

**\*For correspondence:**
tmignot@imm.cnrs.fr

## Introduction

Bacterial biofilms and microbiomes are now under intense study due to their importance in health and environmental issues. Within these spatially structured communities, analysis of cell-cell interactions requires powerful descriptive tools to link molecular mechanisms in single cells to cellular processes at community scales. Microscopy-based imaging methods combining multiple imaging modalities (e.g. bright-field, phase-contrast microscopy, fluorescence microscopy) directly record morphological, spatio-temporal, and intracellular molecular data in a single experiment. However, extraction of quantitative high-resolution information at high-throughput requires accurate computational tools to detect bacterial cells and correctly assign analyzed properties to these cells. While this task might seem trivial to the naked eye, it is a significant computational challenge to detect bacterial cells in a 2D image with high accuracy in dense microcolonies where the cells are in tight contact and under various imaging modalities.

Semantic segmentation of an image assigns pixels to the object that it belongs to, for example bacterial cells or background (*Jeckel and Drescher, 2021*). Traditionally in an image, pixel intensities that exceed a given threshold are assigned to detected objects while pixels with intensities lower than the threshold are assigned to background, producing so-called segmented masks. Detecting bacterial cells among the objects can be obtained in a number of ways using morphometric procedures.

This is a fast moving field and an extensive list of available methods is described by *Jeckel and Drescher, 2021*. In 2D, broadly used example methods such as MicrobeJ (*Ducret et al., 2016*) and Oufti (*Paintdakhi et al., 2016*) use characteristic morphometric parameters (length, area, circularity, feature detection (ie septa) etc...) to filter non-cell objects from the segmented image and fit the remaining objects to embedded cell models, allowing cell pole detection, septa detection, protein localization etc… While these methods are highly performant to study isolated bacterial cells, they are ill-suited to perform segmentation of dense bacterial communities, mostly because adjacent cells are not easily resolved by intensity-thresholding when bacteria are in tight contact due to lower contrast at the interior of the colony (the so-called shade-off artifact). At best, when performed with Oufti, segmentation of single bacteria within micro-colonies requires extensive manual-tuning of multiple parameters limiting its robustness for high-throughput automatic data extraction (*Paintdakhi et al., 2016*; *Stylianidou et al., 2016*).

Machine-learning-based techniques are powerful alternatives to overcome the above limitations of traditional segmentation approaches. For example, one of them, Supersegger, combines intensity-based thresholding with supervised cell boundary recognition on phase contrast images, successfully resolving individuals cells at low contrast at the colony interior (*Stylianidou et al., 2016*). This method is, however, limited to phase contrast images and requires a number of filter applications (ie thresholding, contrast filter and watershed) to appropriately segment these images. Such tuning which may need to be adjusted for each field of view (in addition to image intensity normalization) renders Supersegger difficult to use for the segmentation of large colonies of various bacterial species captured under various imaging modalities. Deep-learning algorithms called Convolutional Neural-Networks (CNNs) have recently shown great promise for image classification and in particular semantic segmentation with reasonable computational power (*Van Valen et al., 2016*). Van Valen and collaborators (*Van Valen et al., 2016*) provided a general proof of concept that CNNs (DeepCell) could be used to segment both eukaryotic and bacterial cells in dense contexts with limited training datasets. However, while the study provides important tips toward the successful training of a CNN for a specialized cell segmentation task, it does not provide a trained CNN for the general segmentation of dense bacterial communities. Nevertheless, the approach provided promising perspectives to segment *E. coli* microcolonies on agar and confirming this, a CNN-based tool (DeLTA, *Lugagne et al., 2020*) was recently developed to detect and track *E. coli* cells immobilized in microfluidic devices (the so-called mother machine, *Lugagne et al., 2020*). Inspired by these methods, we decided to develop MiSiC (*Mi*crobial *S*egmentation *i*n dense *C*olonies), a general CNN-based tool to segment bacteria in single or multispecies 2D bacterial communities at very high throughput. MiSiC is based on U-net, a CNN encoder-decoder architecture that has previously been applied for detection and counting of eukaryotic cells (*Falk et al., 2019*), that relies on shape rather than intensity information and thus performs semantic segmentation of microbial colonies under any microscope modality. Thus, contrarily to most other softwares, MiSiC is insensitive to specific imaging conditions which often require tailored training data sets. Operating MiSiC requires minimal parameter tuning and can be run with standard computational power, in a Napari Graphic User Interface (GUI, a complete user handbook for installation and use is provided) under Python (https://napari.org/). Both semantic segmentation and instance segmentation (in which each cell is defined as a distinct object [*Jeckel and Drescher, 2021*]) masks can thus be easily obtained with minimal computational expertise, making MiSiC broadly applicable to the field of bacterial cell biology.

## Results

### The MiSiC workflow

We sought to develop a prediction workflow that converts an input image taken under phase contrast, fluorescence, or bright field into a binary mask for cell bodies. However, when microscopy is performed under different modalities, pixel intensity variations between imaging conditions make it difficult to perform semantic segmentation with a single procedure. Thus, to minimize the impact of image intensity fluctuations that inevitably arise from varying imaging sources, the input images were transformed into intermediate image representations obtained from the shape and curvature (the Hessian or second-order differentiation) of the imaged objects. This strategy is possible because in rod-shaped bacteria, the characteristic dome-shaped curvature of the poles is remarkably conserved

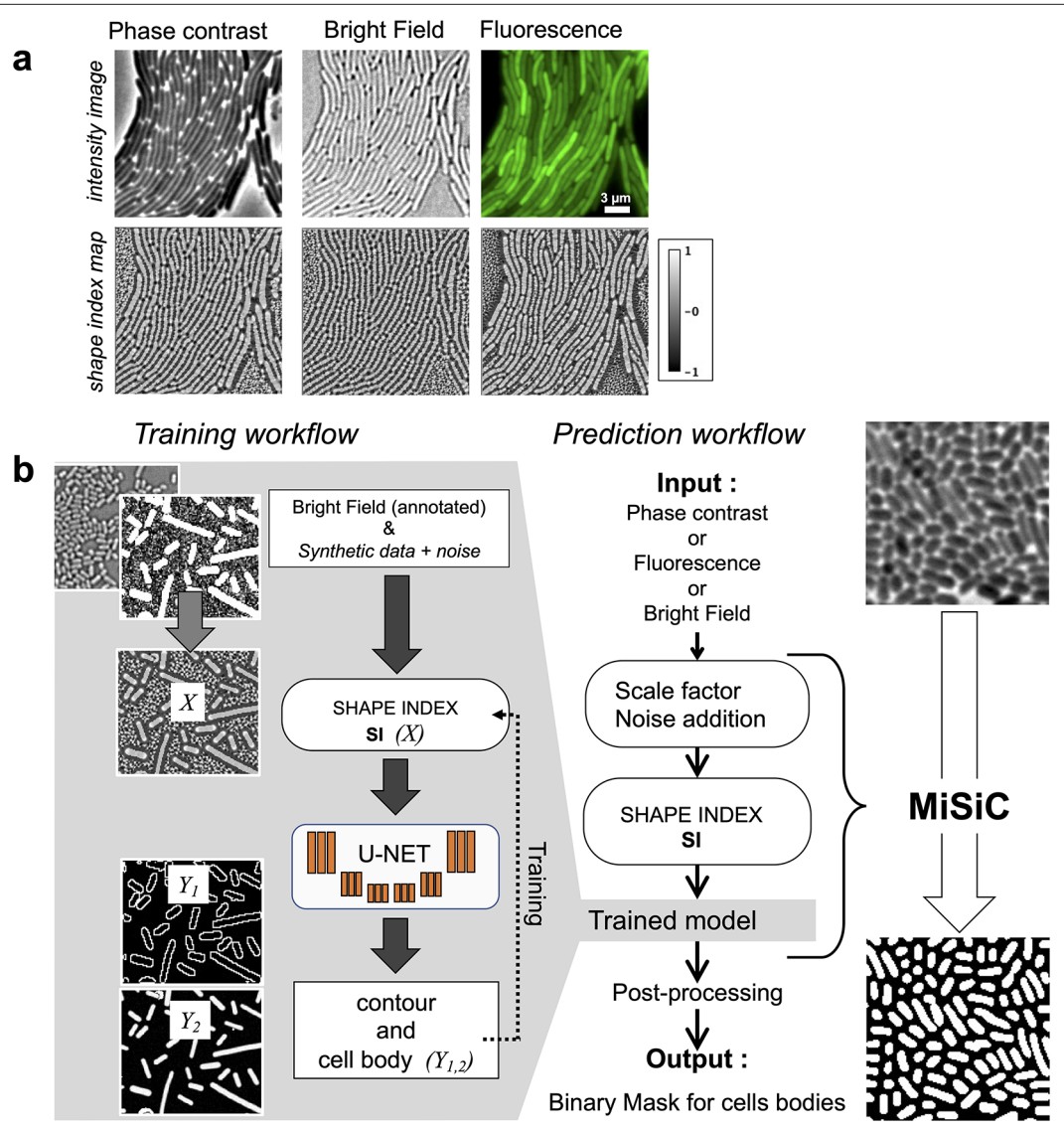

**Figure 1.** MiSiC: A U-netbased bacteria segmentation tool. (**a**) Examples of Shape index maps (SI) calculated from Phase Contrast, Bright Field and Fluorescence images of the same field of *Myxococcus xanthus* cells. (**b**) A set of annotated bright-field images of *Escherichia coli and Myxococcus xanthus* along with synthetic labeled data with additive Gaussian noise was used to generate a training dataset of input images, X, consisting of Shape Index Map of intensity images (at three scales) and segmented images, Y, consisting of contours (**Y1**) and cell body (**Y2**). A CNN with U-net architecture was trained to segment the Shape Index Maps into cell body and contour of bacterial cells. Prediction using MiSiC requires that the mean width of the bacteria in the input image is close to 10 pixels, which is easily obtained by rescaling the input image based on the average width of the bacteria under consideration. Gaussian noise may be added to the input image to reduce false positives (Materials and methods).

The online version of this article includes the following figure supplement(s) for figure 1:

**Figure supplement 1.** Background noise can lead to spurious cell detection by MiSiC.

across division cycles (*Campos et al., 2014*). The curvature changes in the intensity field of an image are thus represented in a so-called Shape Index map (SI) derived from the eigenvalues of the Hessian of the image (*Koenderink and van Doorn, 1992*) (see Materials and methods section). Therefore, all microscopy images can be transformed into SI images with pixel intensity values varying as a function of object curvature and ranging from −1 to +1, a −1 value representing a negative dome-shape and +1 representing a positive dome-shape (*Koenderink and van Doorn, 1992*, *Figure 1A*). SI images of bacterial cells acquired under various experimental conditions were used to train a U-Net.

U-net type architectures allow fast learning from a relatively small body of labeled data because the embedded skip-connections allow the convolutional kernels between both encoder and decoder ends to be shared (*Falk et al., 2019*). Nevertheless, the labeled data must be representative of the broadly varying experimental conditions to produce reliable outputs: in our case, different bacterial species recorded under varying imaging modalities. A schematic of the training strategy is shown in *Figure 1* and detailed in the Materials and methods section. Specifically, the U-Net was trained to segment cells by learning shapes of individual bacteria and patterns emerging from the tight contact between cells. Accordingly, we curated a hand-segmented dataset of 350 bright-field images (n = 34807 cells) of two rod-shaped bacterial species, *Escherichia coli* and *Myxococcus xanthus*. This annotated data was however insufficient for the robust segmentation of bacterial cells, thus it was further enriched with synthetic representations of rod-shaped bacteria of varying length but fixed 10 pixels width corresponding to ≈ 0.6 µm. Overall, the ground truth data used for training had two classes: one corresponding to bacteria cell bodies and the other corresponding to the contour of the detected cell (*Figure 1B*). This makes post-processing possible when there is not enough edge information for the proper separation of tightly connected bacterial cells, using algorithms like watershed, conditional random fields, or snake segmentation (*Yang and Cao, 2013*, see below).

Prior to segmentation, two parameters must be adjusted to generate a SI image from an input image. Because the width of bacterial cells was set to 10 pixels in the training data set, the input image must be scaled similarly so that the width of bacterial cells is also contained in 10 pixels. However, this scaling often smoothens the original image, which in turns smoothens the corresponding SI Image (*Figure 1—figure supplement 1*). This is potentially problematic because the U-Net distinguishes smooth curvatures with well-defined boundaries and noise reduction can lead to increased false positive segmentation in the scaled images. Such false positive detection is more frequent in images where the number of bacteria is sparse. This problem can be solved by adding synthetic noise to the scaled images. In total, the MiSiC workflow takes raw input images of any imaging modality, scales them and adds noise to generate SI that are then segmented with the above described U-Net (*Figure 1B*). For the users, scaling and potential addition of noise can be easily adjusted in the Napari GUI, which is explained in detail in a dedicated handbook.

## MiSiC can segment bacteria under varying microscope modalities

We first tested whether MiSiC can efficiently segment images of bacteria of distinct shapes (*E. coli*, smaller and thicker cells and *M. xanthus*, longer and thinner cells) in dense colonies, captured by phase contrast, fluorescence or brightfield. To quantify the accuracy of segmentation and compare the relative performance of MiSiC for each modality, we compared the obtained MiSiC masks with hand-annotated masks (considered ground truth masks) of the same images and measured the Jaccard index (JI) as a function of the Intersection-over-Union (IoU) threshold for each modality (see Materials and methods, *Figure 2a and b*, *Figure 1—figure supplement 1* a,b). In all cases, high JI values ( ≥ 0.8) were observed for IoU thresholds 0–0.6, indicating that MiSiC can robustly segment all modalities. In fact, MiSiC performance was comparable to the observer's eye for each modality (or even slightly better) because similar JI scores were obtained when the same ground-truth data (generated by the same observer consistently throughout the study) was compared to data annotated by another independent observer (*Figure 1—figure supplement 1c*). As would be expected, the quality of the segmentation was nevertheless variable (albeit slightly) as a function of the imaging modality, the best results being obtained for fluorescence (if the fluorescence is homogenous between cells as seen in *E. coli*, less so for *Myxococcus* cells where the fluorescence levels were more variable), followed by phase contrast and bright field.

## MiSiC can segment colonies of various species and is tolerant to shape deviations

We next tested if MiSiC can segment bacterial species other than those that it was trained on (*E. coli* and *M. xanthus*), other rod shapes (*Pseudomonas aeruginosa*), curved shapes (*Caulobacter crescentus*) and filamentous shapes (*Bacillus subtilis*) (*Figure 3a and b*). For each analysis, we derived JI scores based on the comparison between MiSiC masks and hand-annotated data as described above. MiSiC predicted accurate masks for each bacterial species with JI scores ≥ 0.8 up to 0.5 IoU thresholds for all cases. Nevertheless, and as expected the segmentation accuracy was lower for curved bacteria

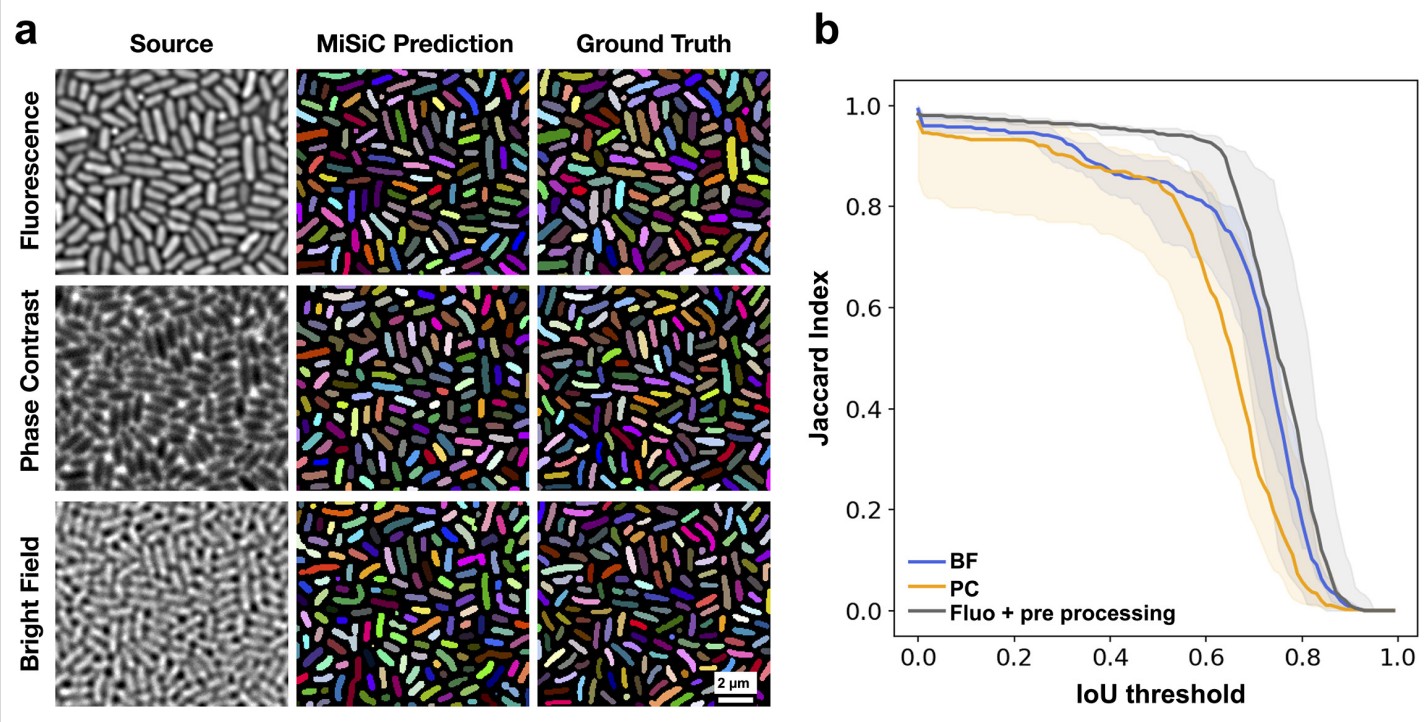

**Figure 2.** MiSiC predictions under various imaging modalities. (**a**) MiSiC masks and corresponding annotated masks of fluorescence, phase contrast and bright field images of a dense *E. coli* microcolony. (**b**) Jaccard index as a function of IoU threshold for each modality determined by comparing the MiSiC masks to the ground truth (see Materials and methods). The obtained Jaccard score curves are the average of analyses conducted over three biological replicates and n = 763, 811, 799 total cells for Fluorescence, Phase Contrast and Bright Field, respectively (bands are the maximum range, the solid line is the median). The fluorescence images were pre-processed using a Gaussian of Laplacian filter to improve MiSiC prediction (describe in Materials and methods).

The online version of this article includes the following source data and figure supplement(s) for figure 2:

**Figure supplement 1.** MiSiC predictions under various imaging modalities.

**Source data 1.** *E. coli* IoU over threshold -Brightfield.

**Source data 2.** *E. coli* IoU over threshold -Fluorescence.

**Source data 3.** *E. coli* IoU over threshold – Phase contrast.

**Source data 4.** Thresholds.

**Figure supplement 1—source data 1.** *Myxococcus* IoU over threshold -Brightfield for panel b.

**Figure supplement 1—source data 2.** *Myxococcus* IoU over threshold -Phase Contrast for panel b.

**Figure supplement 1—source data 3.** *Myxococcus* IoU over threshold-Fluorescence for panel b.

**Figure supplement 1—source data 4.** Threshold for panel 1 c.

**Figure supplement 1—source data 5.** Ground truth IoU over threshold-Brightfield for panel c.

**Figure supplement 1—source data 6.** Ground truth IoU over threshold-Phase contrast for panel c.

**Figure supplement 1—source data 7.** Ground truth IoU over threshold-Fluorescence for panel c.

**Figure supplement 1—source data 8.** Observer IoU over threshold-Brightfield for panel c.

**Figure supplement 1—source data 9.** Observer IoU over threshold-Phase contrast for panel c.

**Figure supplement 1—source data 10.** Observer IoU over threshold-Fluorescence for panel c.

than rod shaped bacteria (compare *P. aeruginosa* and *C. crescentus*). For filamentous bacteria, the filaments were remarkably well resolved but that was not always the case for cell separations (septa) within the filaments, likely because of insufficient edge information at cell septa in the raw image. Note that this problem may be resolved by post-processing, for example by applying a watershed algorithm to the MiSiC mask (*Figure 2—figure supplement 1*), which effectively resolves additional undetected septa. Although MiSiC can thus detect a number of bacterial species, a current limitation

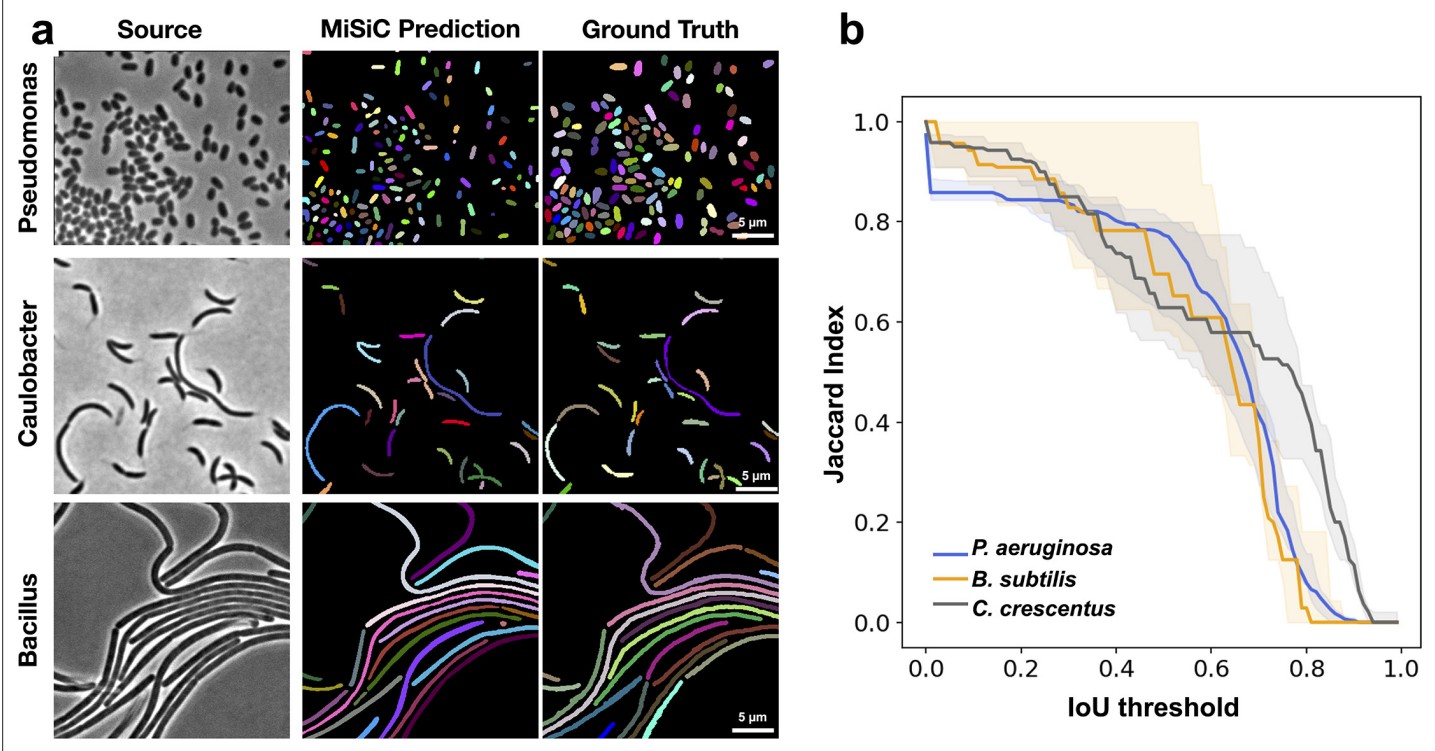

**Figure 3.** MiSiC predictions in various bacterial species and shapes. (**a**) MiSiC masks and corresponding annotated masks of phase contrast images of another *Pseudomonas aeruginosa* (rod-shape), *Caulobacter crescentus* (crescent shape) and *Bacillus subtilis* (filamentous shape). (**b**) Jaccard index as a function of IoU threshold for each species determined by comparing the MiSiC masks to the ground truth (see Materials and methods). The obtained Jaccard score curves are the average of analyses conducted over three biological replicates and n = 1149,101, 216 total cells for *P. aeruginosa*, *B. subtilis* and *C. crescentus*, respectively (bands are the maximum range, solid line the median). Note that the *B. subtilis* filaments are well predicted but edge information is missing for optimal detection of the cell separations.

The online version of this article includes the following figure supplement(s) for figure 3:

**Figure supplement 1.** Refining cell separations with watershed.

**Source data 1.** IoU over threshold -*Bacillus subtilis*.

**Source data 2.** IoU over threshold -*Caulobacter crescentus*.

**Source data 3.** IoU over threshold – *Pseudomonas aeruginosa*.

lies in its ability to segment round or oval cells. In fact, such shapes were initially omitted from the original training dataset to omit the spurious detection of round non-cell objects that are frequently observed in background images.

## MiSiC as a tool to study dynamic bacterial communities

Encouraged by these results, we tested whether MiSiC could be further used to study bacterial multicellular organization and inter-species interactions accurately at very large scales. As a model system we used *Myxococcus xanthus,* a delta proteobacterium living in soil, that predates collectively in a process whereby thousands of cells move together to invade and kill prey colonies (*Figure 4*, *Pérez et al., 2016*). In the laboratory, spotting a *Myxococcus* colony next to a prey colony (here *E. coli*) results in invasion and complete digestion of prey cells in 48 H (*Figure 4b*). To capture predator-prey interactions at single cell resolution, we set up a predation assay where *Myxococcus* and *E. coli* interact on a 1 cm² agar surface directly on a microscope slide (*Figure 4—figure supplement 1a*). Under these conditions, the entire invasion process occurs over a single prey cell layer allowing identification of single predator and prey cells at any given stage. This area is nevertheless quite large, and to record it with cell-level resolution, we implemented a multi-modal imaging technique termed 'Bacto-Hubble' (in reference to the Hubble telescope and its use for the reconstruction of large scale images of the galaxies) that scans the entire bacterial community with a 100 X microscope objective

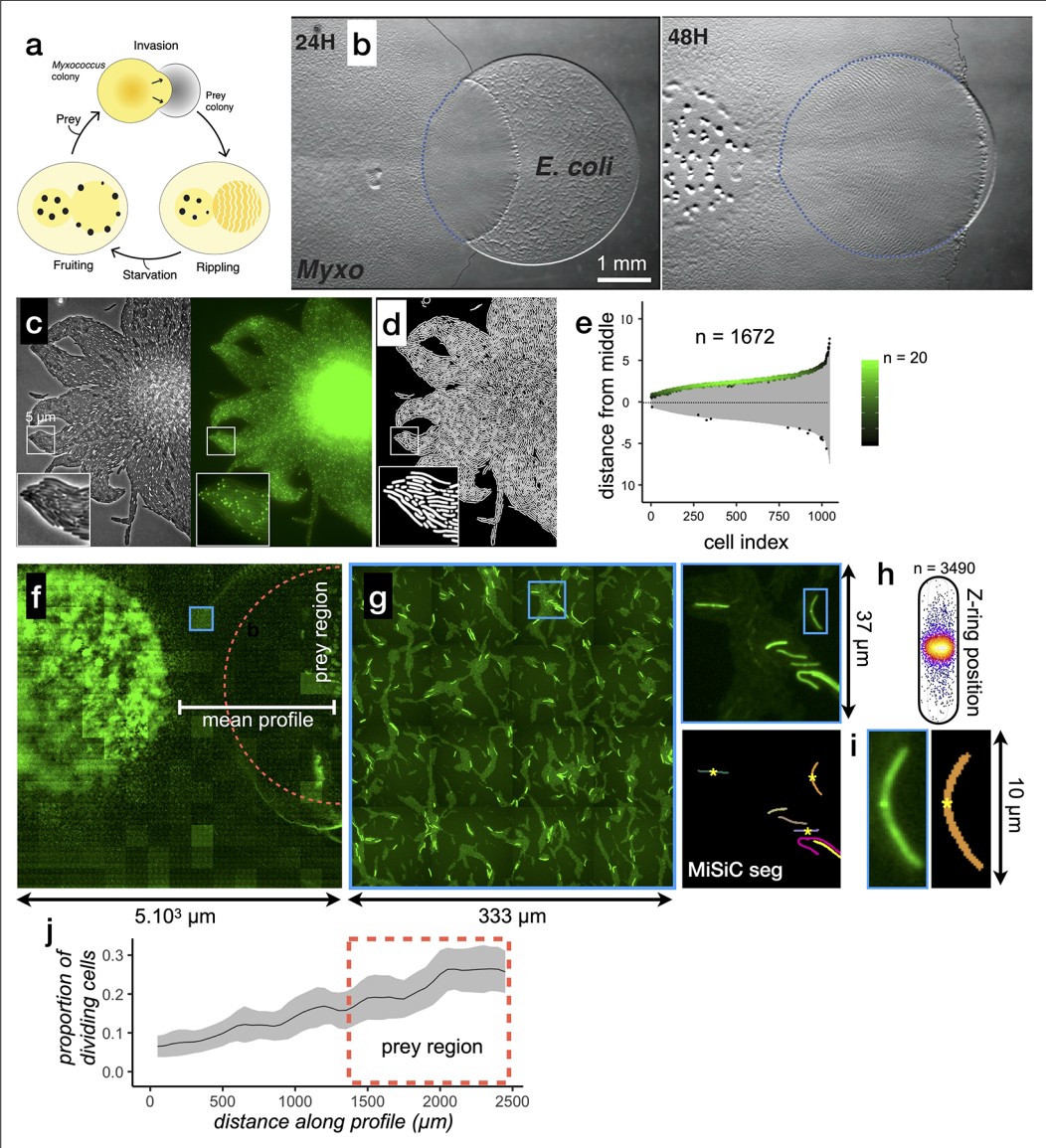

**Figure 4.** MiSiC can be applied to the study of cellular processes at the mesoscale. (**a**) The *Myxococcus xanthus* predation cycle. *Myxococcus* cells use motility (yellow colony) to form so-called rippling waves to invade and consume prey colonies (gray). When encountering starvation (outside the prey) or after the prey is consumed, the Myxcococcus cells form aggregate that matures into fruiting bodies (black dots) where the cells differentiate into spores. These spores remain dormant until another prey is encountered which provokes the cycle to resume (see *Pérez et al., 2016* for details about the cycle).(**b**) The *Myxococcus xanthus* cycle can be observed directly on a Petri dish. Shown are 24 H and 48 H time points. At 48 H, spore-filled fruiting bodies are observed forming in the nutrient depleted area but not in the former prey area where the *Myxococcus* cells are actively growing. This stage corresponds to the rippling stage shown in (**a**). Scale bar = 0.5 mm. (**c–e**) MiSiC can segment dense bacterial swarms.(c) An *M. xanthus* swarm expressing SgmX-GFP, observed at colony edges and captured under phase contrast, fluorescence and corresponding magnified images.(**d**) MiSiC prediction mask obtained on the phase contrast image shown in b (**e**) Demograph representation of the segmented cells in the MiSiC mask (**d**) and corresponding localization of the SgmX-GFP fluorescent clusters. The horizontal axis represents the number of cells ordered by cell length. The vertical axis represents cell coordinates in µm and aligned such that polar fluorescence clusters have positive values (with respect to the position of the cell middle set to 0). The color of the clusters reflects the number of cells for each given cell length (bins = 0.05 µm, maximum = 20 cells for [3.9–3.95] µm). n = 1672 detected cells after filtering the MiSiC mask. (**f–j**) Mapping of *M. xanthus* cell division in the *M. xanthus-E. coli* community. (**f–g**) Bacto-Hubble image of a predatory field containing FtsZ-NG labeled *Myxococcus xanthus* cells and unlabeled *Escherichia coli* prey cells. The composite image results from the

*Figure 4 continued on next page*

*Figure 4 continued*

assembly of 15 × 15 Tile images. The dotted circle marks the limits of the original prey colony. The white line (mean profile) indicates the axis used for the analysis shown in (**i**). (**f**), a single image tile showing a representative density of fluorescent cells. (**h**) Detection of dividing cells. FtsZ-NG fluorescent clusters are detected at midcell. The FtsZ clusters can be detected as fluorescence intensity maxima. Shown is a projection of the position of fluorescence intensity on a mean cell contour for a subset of n = 3490 cells (representing 14 % of the total detected cells with a cluster), revealing that as expected, the clusters form at mid-cell. The blue square marks the cell shown as an example in (**h**). (**i**) Counting dividing cells. The example shows segmentation of the field shown in (**h**). The position of Z-ring foci detected as fluorescent maxima was linked to all fluorescent cells segmented in the MiSiC mask. (**j**) *Myxococcus* cells divide in the prey colony. The spatial density of dividing cells, the total fluorescent cells (Materials and methods) and the proportion of dividing cells (density of dividing cells/density of total cells, Materials and methods) were determined all across the prey area shown in (**e**) (dotted circle). The mean ratio and standard deviation are plotted along a spatial axis (distance along profile) corresponding to areas outside and inside (dotted rectangle) of the prey area (mean profile, white segment in (**e**)).

The online version of this article includes the following source data, source code, and figure supplement(s) for figure 4:

**Source code 1.** Demograph in R for *Figure 4e*.

**Source data 1.** Fluorescence maxima – *Figure 4e*.

**Source data 2.** Cell length – *Figure 4e*.

**Source data 3.** Z-ring positioning for *Figure 4h*.

**Figure supplement 1.** Bacto-Hubble captures millimeter size images of bacterial communities which can be segmented with MiSiC.

**Figure supplement 1—source data 1.** DICE index as a function of noise for panel d.

and reconstructs a single image by near neighbor end-joining of multiple tiles of 80 nm/pixel resolution images (*Figure 4—figure supplement 1b*). Application of this method requires addressing practical considerations that are detailed in the methods section. Bacto-Hubble images (phase contrast and multi-channel fluorescence) thus capture cellular processes in a native community environment. We next tested whether MiSiC addresses the computational challenges posed by the analysis of such complex (dense population and mixed species) and large size data sets.

First, we tested the capacity of MiSiC to segment closely-packed swarms of *Myxococcus xanthus* cells captured in a single image tile. To test the fidelity of segmentations in these conditions, we imaged a swarm composed of cells expressing SgmX-sfGFP, a motility protein that localizes to the cell pole (*Mercier et al., 2020*; *Potapova et al., 2020*) in both fluorescence and phase contrast modalities (*Figure 4c*). Phase contrast images were used to obtain a MiSiC segmentation mask (*Figure 4d*). Subsequently, the mask was processed under MicrobeJ (*Ducret et al., 2016*) to remove objects that do not correspond to cells (Materials and methods, less than 1.4 %, n = 1695) and calculate the localization pattern of SgmX-GFP foci with respect to the long axis of each segmented cell (*Figure 3e*). As expected, SgmX-GFP loci localized to a cell pole, consistent with most *Myxococcus* cells in swarms being properly segmented by MiSiC.

Second, to show that MiSiC can be used to quantitatively study cellular processes in entire Bacto-Hubble images, we mapped a *Myxococcus* cellular process directly during prey invasion. Cell division is expected to occur mostly in prey-areas in absence of any other source of nutrients. Like all rod-shaped bacteria, dividing *Myxococcus* cells assemble a polymeric FtsZ bacterial tubulin ring to initiate cell division (*Schumacher et al., 2017*). When it is fused to fluorescent proteins, the FtsZ ring is observed as a dot at mid-cell (*Treuner-Lange et al., 2013*), which can be used as a proxy to determine which cells enter division. Thus, we first engineered *M. xanthus* cells expressing FtsZ fused to Neon-Green (NG, Materials and methods) and mixed *Myxococcus* FtsZ-NG+ (5%) with non-labeled cells (95%) in the presence of an *Escherichia coli* prey cell colony. A fluorescence Bacto-Hubble image spanning ~5 mm$^2$ (representing 225 tiles of 500 × 500 pixels images) of the community during the invasion phase (*Figure 4f*) was then captured and segmented tile-by-tile using MiSiC (*Figure 4f–g*). Due to the fact that MiSiC uses SI, it is remarkably insensitive to noise and intensity variations between images (*Figure 4—figure supplement 1c,d*) and thus it easily allows segmentation of multi-tile images where signal intensity variations between tiles is to a large extent unavoidable. Cells with mid-cell FtsZ-NG fluorescence clusters were clearly observed suggesting that cell division is ongoing

(*Figure 4h*). Dividing cells were therefore counted across the entire image (*Figure 4i*, Materials and methods) to determine where they localize spatially within the community. *Figure 4j*, shows that cell division is markedly increased in the prey area, demonstrating directly that *Myxococcus* grows during prey invasion. Thus, MiSiC is appropriate for the automated detection of cellular processes (detected at subcellular resolution) at community scales.

Third, we explored how MiSiC could be further adapted to segment and classify multiple bacterial species intermingling and interacting in space; here *Myxococcus* cell groups invading the tightly-knitted *E. coli* prey colony. To segment each bacterial species directly from unlabeled phase contrast-images, new training datasets were produced and used to retrain the U-NET. These labeled datasets were obtained by imaging GFP-labeled *Myxococcus* and mCherry-labeled *E. coli* (see Materials and methods). Images were captured for each channel (GFP, mCherry) and segmented separately with MiSiC to obtain masks for each species (*Figure 5a*). These masks were used to retrain the U-NET, the predictive value of which was tested by deriving classification masks directly from a phase-contrast image of a mixed *Myxococcus* (GFP) - *E. coli* (mCherry) population (*Figure 5*). Comparison of the MiSiC classification to hand annotated GFP (*Myxococcus*) and mCherry (*E. coli*) images over selected fields where each species interact (n = 4, ie *Figure 5b*) respectively yielded JI scores of 0.95 ± 0.036 (n = 200 cells) and 0.89 ± 0.047 (n = 545 cells), suggesting that the classification is highly accurate despite the tight interactions between *Myxococcus* and *E. coli* cells. To further show that species classification is reliable in a large dataset, we next classified *Myxococcus* from *E. coli* cells directly in a phase contrast Bacto-Hubble image (840 tiles, $2.1.10^9$ pixels). We could thus discriminate cells belonging to each species in the predation area (*Figure 5c*) for a total detection of ~402,000 *Myxococcus* and ~630,000 *E. coli* cells in the entire image. Given the large size of the dataset, it would be impossible to test the accuracy of the classification procedure exhaustively by comparing it to the ground truth, instead, we tested whether the distribution of shape descriptors, such as the extent (E = area/bounding box area), solidity (S = area/convex area) and minor axis length, matched the distribution of these descriptors obtained from images of each single bacterial species (*Figure 5d*). The observed distributions were indeed consistent with an efficient separation of species in the mixed community, suggesting that the classification is also robust in large images. Inevitably, infrequent ambiguous classifications arise in the areas where the cells interact tightly due to low contrast in these areas. These instances generally appear as bi-color objects because the prediction is not homogeneous inside such objects. Given that *Myxococcus* cells and *E. coli* cells have clear morphometric differences, these uncertain cells can be easily eliminated by filtering with discriminating parameters (i.e. each of them or combined, *Figure 5d*).

## Discussion

In this article, we have presented MiSiC, a deep-learning based bacteria segmentation tool capable of segmenting bacteria in dense colonies imaged through different imaging modalities. The main novelty of our method is the use of a shape index map (SI) as a preprocessing step before network training and segmentation. The SI depends on the Hessian of the image, thus preserving the shape of bacterial masks rather than the raw intensity values, which vary as a function of microscopy modalities. In general, the use of SIs rather than image intensity is a promising lead for any deep learning approach to cell segmentation that relies on shape, which could also solve modality issues for eukaryotic cell and cellular organelle segmentation. Another important asset of MiSiC is the use of synthetic data to enrich training data sets, greatly facilitating ground truth annotations. Combined with the use of SIs, we show that this strategy only requires two adjustment parameters (scaling and noise addition) and it makes segmentation agnostic to imaging modality, and adapted to different bacterial species with different morphologies, provided that they do not deviate too largely from rod shapes. We have currently omitted detection of oval or round cells, to avoid false positive detection of round objects, but in theory, MiSiC could be adapted to such application by simple retraining with an adapted model using both real images and synthetic data. Currently, such retraining requires computational expertise but it is imaginable that a future version of MiSiC would include a GUI to generate training data depending on the needs of the community.

MiSiC is appropriate for the automated analysis of complex images, such as fluorescence (Bacto-Hubble) images tiles. In our hands, FtsZ-NG-expressing cells could not be properly segmented across tiles with intensity-based methods. In fact, due to the large size of the sample and probable issues

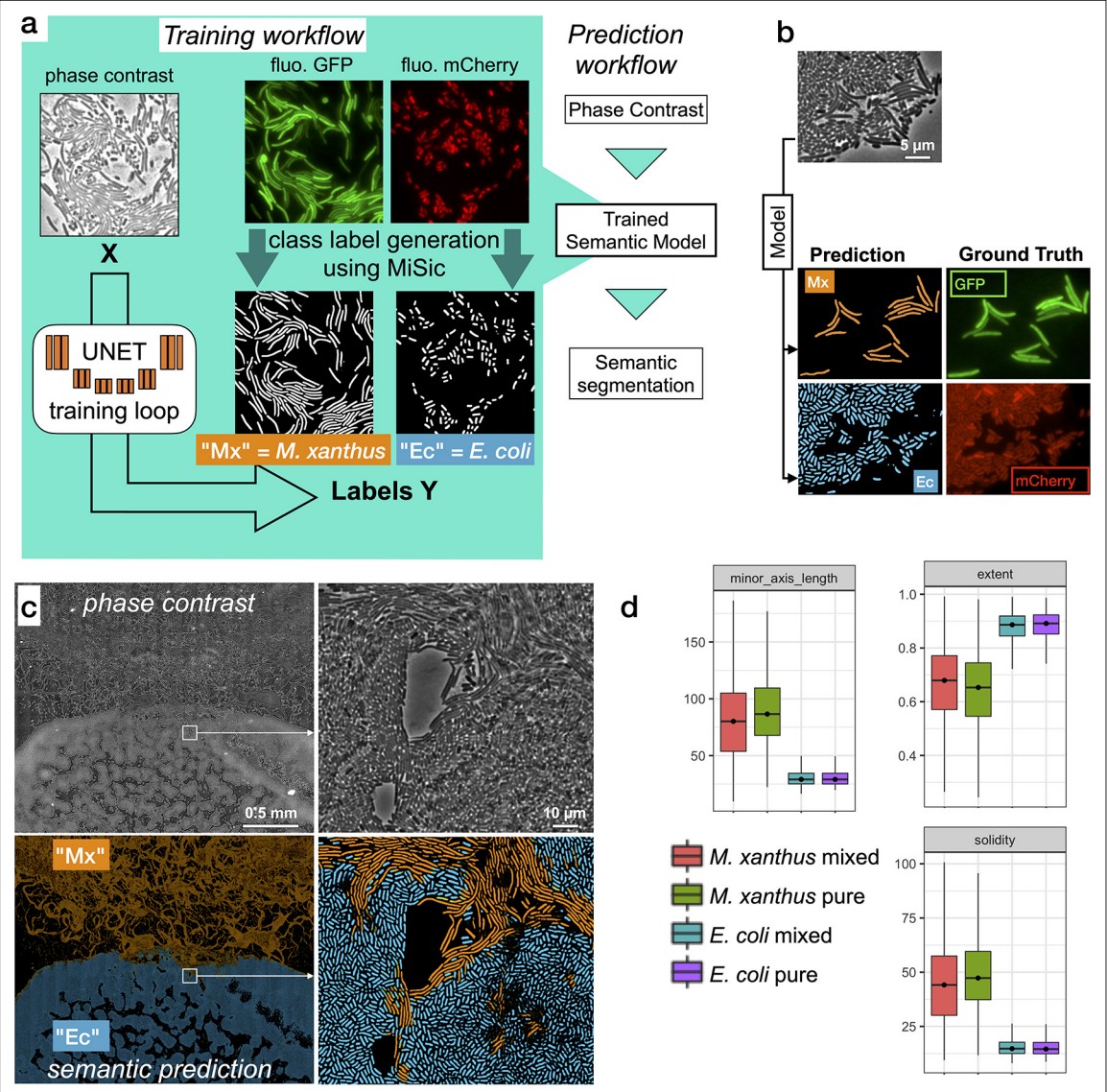

**Figure 5.** Semantic segmentation of *M. xanthus* and *E. coli* from single phase contrast images. (**a**) Semantic classification network. A U-Net was trained to discriminate *M. xanthus* cells from *E. coli* cells. The training dataset consisted of GFP+ *M. xanthus* cells mixed with *mCherry+ E. coli* cells, which were imaged in distinct fluorescent channels (GFP and mCherry) and segmented using MiSiC to produce ground truth data for each species. The network uses unlabeled phase contrast images as input (**X**) and produces one output for each labeled species (Y, Mx or Ec). (**b**) Semantic classification of intermixed *Myxococcus* and *E. coli* cells. Shown is a mixed population of GFP+ *Myxococcus* and mCherry+ *E. coli* cells. The classification (*Myxococcus*, Mx) and *E. coli* (Ec) were obtained directly from the phase contrast image. Corresponding fluorescence images of GFP+ and mCherry+ are shown for comparison and were used to estimate the accuracy of the classification. (**c–d**) Direct semantic classification of *M. xanthus* interacting with *E. coli* in a Bacto-Hubble image. (**c**) Bacto-Hubble image of *M. xanthus* cells invading an *E. coli* colony after 24 hours. The composite image corresponds to 20 × 42 image tiles captured by phase contrast and segmented tile-by-tile to produce the resulting classification. Inset: zoom of an area where *M. xanthus* cells interact tightly with *E. coli* cells within the *E. coli* colony. Phase Contrast and corresponding predictions (*M. xanthus* in orange and *E. coli* in blue) are shown. (**d**) Morphological analyses of the classified cell population and comparison with the ground truth data. Morphological parameters (Extent, Solidity and minor axis length) were determined for the cells predicted in the *M. xanthus* (Mx mixed) and *E. coli* masks (*E. coli* mixed) in the context of a mixed colony and compared to the same parameters obtained from MiSiC segmented from images of pure cultures (Mx/*E. coli* pure).

The online version of this article includes the following source code and figure supplement(s) for figure 5:

**Source code 1.** Boxplots in R for *Figure 5d*.

**Source data 1.** *E. coli* only for *Figure 5d*.

**Source data 2.** *E. coli* in mixed colony for *Figure 5d*.

**Source data 3.** *M.xanthus* only for *Figure 5d*.

**Source data 4.** *M.xanthus* in mixed colony for *Figure 5d*.

with its overall flatness, fluorescence intensities vary across tiles, which required parameter adjustment for each tile, making the task overly complex. These problems are solved in MiSiC because image tiles conversion into SIs cancels fluorescence intensity fluctuations and MiSiC is quite robust to noise (*Figure 4—figure supplement 1c*,d). Tile-by-tile segmentation in MiSiC can thus also allow extraction of high-resolution information from large size data sets while allowing prediction of large images with no exceptional memory requirements Typically, a large data set containing up to 3,000 tiles of 500 × 500 pixels ($7.5.10^9$ pixels) may be segmented in ≈ 50 minutes on a GPU-equipped (Quadro P5000) computer (RAM 64 Gb, Intel Xeon 2.10 GHz). In addition, the Napari environment provides a simple GUI, allowing the generation of MiSiC masks from any image by any user without the need of computational expertise.

MiSiC is therefore broadly applicable to diverse applications in bacterial cell biology and generates prediction masks that can be used with other available softwares. For example, Supersegger which corrects improper segmentation based on cell contours may likely be used to correct aberrant segmentations in a MiSiC mask, combining the advantage of SI-based segmentation and contour correction. Along similar lines, non-cell or improperly separated cell objects an appear in the MiSiC masks and while some can be removed by the introduction of noise, an easy way to do it is to apply a post-processing filter using morphometric parameters to remove objects that are not bacteria. This can be easily done using Fiji, MicrobeJ or Oufti. MicrobeJ also contains a GUI and associated toolbox to cure improper cell contours manually. MicrobeJ and Oufti are especially useful from downstream analyses of the masks because they both allow cell tracking as well as determining protein localization at sub-diffraction resolution.

Finally, we show that MiSiC can be further implemented for the semantic classification of bacterial cell types directly from phase contrast images. While the network developed herein works specifically for *M. xanthus* and *E. coli* discrimination, it provides a proof of concept that the approach can be extended to segment and classify any number of bacterial species provided that a dataset (i.e. fluorescence labeling) is available to train a U-Net with ground truth generated in MiSiC. Given that these tasks must be tailored to specific applications, we did not implement them in this package but in principle, adapting MiSiC for such applications could be developed by any laboratory without deep expertise of CNNs. This is an exciting prospect at a time where tremendous efforts are injected to reconstruct micro communities in synthetic contexts for mechanistic studies (*Wolfe et al., 2014*) . Although MiSiC is designed for the analysis of bacteria that develop in 2D, recent toolboxes for the study of micro communities in 3D are also emerging (*Hartmann et al., 2021*; *Zhang et al., 2020*); we therefore foresee that quantitative microscopy approaches will profoundly impact microbiome research of health and environmental significance.

# Materials and methods
## Bacterial strains and predation assays

The complete list of the strains used for the study is compiled in Table S1 *Supplementary file 1*. For predation assays, cells of *Myxococcus xanthus* (DZ2, Table S1, *Supplementary file 1*) were grown overnight in 100 mL flasks in 10–20 mL of CYE (*Bustamante et al., 2004*) media without antibiotics at 32 °C with shaking. In parallel a colony of *Escherichia coli* (MG1655, Table S1) was grown in 5 mL LB medium in a glass tube at 37 °C with shaking. The next day, $OD_{600\,nm}$ were measured and cultures of both strains were washed twice at 5000 rpm in CF (*Bustamante et al., 2004*) minimal media to discard CYE and LB traces. After the washes, the density of the cultures was brought to 5 OD units in CF media. Pads of CF agar 0.5 % were poured in precast frames (in situ GeneFrame, 65 µL, ABGene, AB-0577) that were mounted on glass slides and briefly dried.  One µL of both *Myxococcus xanthus* and prey cell suspensions were spotted as close as possible to one another on the pad making sure that they would not merge. Glass slides were kept in a sealed humid dish for 6, 24, 48, or 72 hr at 32°C . Thirty min before observation, the agar pad around the colonies was cut out and discarded and the pad was sealed with a cover slip and observed by microscopy.

*Bacillus subtilis* strains were grown in LB medium at 37 °C until they reached an $OD_{600\,nm}$ of 0.6, were transferred into wells and covered in a low melting LB-based agarose suspension (2%) before observation. *Pseudomonas aeruginosa* was grown in LB medium and cells were observed at an $OD_{600\,nm}$ of 0.5. *Caulobacter crescentus* was cultured in PYE on the benchtop without shaking for 3–4 days.

Cells were imaged from disrupted fragments of a pellicle biofilm and were transferred from the air-liquid interface to glass slides for imaging (*Marks et al., 2010*). *Anabaena nostoc* was grown in BG11 medium at 30 °C with illumination (40 µE m-2s-1). Finally, *Desulfovibrio vulgaris* cells were grown until mid-exponential phase ($OD_{600\ nm}$ of approximately 0.4–0.5) in LS4D medium supplemented with 1 g/L of yeast extract (LS4D-YE) at 33 °C in an anaerobic chamber (COY Laboratory Products) filled with a 10% $H_2$-90% $N_2$ mixed-gas atmosphere. Cultures (200 µL) were centrifuged, and the pellet was resuspended in 100 µl of 10 mM Tris-HCl (pH 7.6), 8 mM $MgSO_4$ and 1 mM $KH_2PO_4$ buffer (TPM buffer). The cells were placed between a coverslip and an agar pad containing 2 % of agarose.

## Molecular biology and strain construction

To follow cell cycle progression in single cells of *Myxococcus xanthus*, a merodiploid strain of DZ2 expressing both native FtsZ and the fusion protein FtsZ-neonGreen (FtsZ-NG, Table S3 *Supplementary file 3*) was built (DM14, Table S1, *Supplementary file 1*). To do so, the coding sequence of DZ2 *ftsZ* gene (MXAN_5597) along with its predicted promoter sequence was amplified by PCR with primers oDM1 and oDM2 (Table S2, *Supplementary file 2*) and cloned in the non-replicative plasmid pKA32 (*Treuner-Lange et al., 2013*) allowing for its site-specific integration at the DZ2 *attmx8* phage attachment site on the *M. xanthus* chromosome. The coding sequence of the neonGreen protein was amplified from a plasmid (*Shaner et al., 2013*) using primers oDM16 and oDM17 (Table S2 *Supplementary file 2*) allowing the in frame addition of neonGreen at the C-terminus of the FtsZ protein. When grown in CYE rich medium, DM14 did not present any significant defect in growth rate or cell shape. DM14 cell size is not significantly different from that of the isogenic wild type DZ2 strain. DM14 cells were spotted on thin CF agar pads to follow FtsZ localization in axenic cultures and allowed us to confirm that cell cycle progression was accompanied with the relocalization of Ftsz-nG from being diffuse in the cytoplasm to forming a discrete fluorescent focus at mid-cell before cell septation as previously described (*Treuner-Lange et al., 2013*).

To generate Dataset 2 (see below), strains of *E. coli* (EC500, Table S1 *Supplementary file 1*) and *M. xanthus* (DM31, Table S1 *Supplementary file 1*) expressing soluble versions of mCherry and sfGFP fluorophores respectively were used. To generate DM31, a plasmid allowing for the constitutive expression of sfGFP was built (pDM14 Table S3 *Supplementary file 3*). To obtain a high and constitutive expression of sfGFP in *M. xanthus*, we sought for the closest homolog of the constitutively expressed *E. coli* EF-TU (Translation elongation factor) in *M. xanthus* genome which is MXAN_3068. The 1000 bp region upstream of MXAN_3068 (*p3068*) was amplified by PCR using oDM53 and oDM54 and cloned upstream the coding sequence of sfGFP (amplified using primers oDM61 and oDM62) in a pSWU19 plasmid. The transcriptional fusion was then integrated on DZ2 chromosome at the *attmx8* site through transformation. DM31 cells display a constitutive bright diffuse fluorescent signal in our growth conditions.

## Microscopy and image acquisition

All microscopy images were acquired with an inverted optical microscope (Nikon TiE) and a 100 x NA = 1.45 Phase Contrast objective. Camera used was Orca-EM CCD 1000 × 1000 (Hamamatsu) camera mostly set with binning 2 × 2. Acquisition software was Nikon NIS-Elements with specific module JOBS. Fluorescence acquisition used a diode-based excitation device (Spectra-X Lumencore).

The Bacto-Hubble image is a composite image of rasters of the entire area and requires that the scanning speed must be sufficiently fast to avoid image shifts due to ongoing cell dynamics. To minimize the shift in focus from tile to tile we used Nikon perfect focus system (PFS) equipped with servo-control of the focus with an infrared LED. This is especially challenging because continuous focus alignment of the microscope slows down the acquisition times dramatically. To obtain a satisfactory compromise allowing both fast scanning and correct focusing we: (i) reduced the number of dynamic elements on the microscope set up: we replaced shutters by Light Emitting Diodes (LEDs, Spectra-X for fluorescence source and a white diode for transmitted light) which could be switched with a high-frequency rate (100 kHz). In addition, a double band dichroic mirror for the fluorescent cube was used to avoid switching the filters' turret for each snapshot. (ii) used an EM-CCD camera set to a 2 × 2 binning mode to reduce the size of images (500 × 500 pixels at 0.16 µm/px) and acquisition time, and (iii) sped up the vertical movements by means of a piezoelectric stage. In its largest scanning mode, Bacto-Hubble thus captures 80 × 40 raster images covering a total surface of 20 mm2 (containing up

to 0.8 billion pixels, an acquisition-time up to 4 hr), enabling a continuous magnification display from eye visible structures to single-cells. Individual tiles for Bacto-Hubble images were acquired with the scan large field capabilities of NIS-Elements software. A key point for Bacto-Hubble large images was the quality of the sample slide mounting. The samples were placed on a glass slide with a thin double-sided sticky frame (in situ GeneFrame, 65 µL, ABGene, AB-0577). An agar pad was poured inside the frame and a microliter of cells was placed on it. The chamber was closed with a glass cover slide. This assembly allows very good flatness and rigidity.

## U-NET as base architecture

We implement a U-net inspired in encoder-decoder architecture with skip-connections and use it as a base network for segmentation tasks. This architecture is now widely used for segmentation tasks and has many advantages that are discussed in previous articles (*Falk et al., 2019*; *Ronneberger et al., 2015*). The original U-net architecture (*Falk et al., 2019*) was modified to include relu activation for all layers except for the output layer where sigmoid activation was used. The general U-net was implemented in Python programming language using tensorflow (https://www.tensorflow.org/tutorials/images/segmentation), where the number of input channels (say, n) and output classes (say, m ) could be varied as required by different models. The number of encoder layers were fixed to four with filter lengths [32,64,128,256] for the encoder side. The loss function was also modified from the original implementation. A combination binary cross entropy and the *Jaccard, 1912* index was used as the loss function with Adams optimizer (learning rate = 0.001) for the minimization.

For brevity, we denote the network as a mapping between input X and output y as,, where X is a set of images with n channels and y is the output with m classes. Thus given a training data set of $X_{train}$ with sizes (N × S × S× n) and multiclass images $y_{train}$ (of size N × S × S× m), the generalized implementation of U-NET learns to predict the segmented image from unknown images X.

## Training datasets

Two training datasets were used for this study. Dataset one to segment bacterial cells (MiSiC) and Dataset two to classify *Myxococcus* and *E. coli* cells in mixed cultures.

### Dataset 1

Combination of hand-annotated data with synthetic data provided the most accurate segmentation after training. When the network was trained with synthetic data only, segmentation was less performant presumably due to the fact that the synthetic data was generated by randomly throwing cell-shaped objects onto an image, which does not capture the intricate patterns created by cell-cell interactions observed in real images of dense cell populations.

This training dataset consists of three parts:

a. A hand annotated dataset corresponding to: 263(training) +87(test) cases of Bright-field images of *Escherichia coli* and *Myxococcus xanthus* with segmented masks. This data set corresponds to images sizing from 157 × 157 to 217 × 217 pixels and containing a variable number of cells (up to 400 cells per image) for a total number of cells of 34,807 manually drawn cells.
b. 200 null cases that contain background images without bacteria taken from bright-field, fluorescence and phase contrast data.
c. A synthetic data set consisting of 600 cases was generated, 30 % containing 16 cells per image (256 × 256 pixels) for sparse densities and 70 % containing a maximum of 327 cells for high densities per image. Furthermore, 200 null cases, generated by inclusion of random gaussian noise and circular objects, were also included in the dataset. The synthetic data was generated with a simple model for rod-shaped bacteria with a width ranging from 8 to 10 pixels. An overlap threshold of 2 % was used to obtain a dense cell population. The binary mask created was then smoothed with a Gaussian filter and Gaussian noise was added to emulate noise in real images.

The ground truth in this training dataset (denoted as [X',y ]) has two classes: one with the mask of bacteria and the other with the contour of the detected cell. The test set consists of 87 cases of labeled bright-field images unseen by the trained network. The accuracy of the network is calculated over this test set.

## Dataset 2

This dataset was used to train the classification network to discriminate *Myxococcus xanthus* and *Escherichia coli* cells directly from phase contrast images. To construct the data set, we obtained images from predatory interactions of *M. xanthus* and *E. coli*, where *M. xanthus* is tagged with green fluorescence (GFP, DM31, Table S1, **Supplementary file 1**) and *Escherichia coli* with red fluorescence (mCherry, EC500 -Shaner et al. 2004- Table S1). The fluorescence images were then processed with a gamma adjustment and segmented using MiSiC to produce clean masks and contours of two classes, namely, *Myxococcus xanthus* and *Escherichia coli*. Thus, y in dataset two contains two channels corresponding to a mask of each class. A total of 4,000 such pseudo-annotated images of size (256 × 256) were used for dataset 2. These images were obtained after random cropping of microscope images (1000 × 1000) and the number of cells varied from image to image.

## MiSiC, shape-index map-based segmentation

$\sigma$ The shape index (SI) map of an image, x, calculated over a scale , is defined as

$$SI\left(x, \sigma\right) = \frac{2}{\pi} tan^{-1}\left(\frac{k_2 + k_1}{k_2 - k_1}\right),$$

where $k_1$, $k_2$ (with $k_1 > k_2$) are the eigenvalues of the Hessian of the image, $x$, calculated over a scale $\sigma$ (**Koenderink and van Doorn, 1992**). The SI map remains within the range [−1, 1] and preserves the MiSiC shape information while being independent of the intensity values of the original image. Using the Dataset 1, we pre-process the input images X' to generate a train set: $X_{train}$ of size 1260 × 256 × 256 × 3. Each channel in $X_{train}$ is the shape-index map calculated at three different scales [1,1.5,2] to obtain shape index information of the cells at various scales. An instance of the U-net is trained over this data set to produce a network able to map data represented by $X_{train} \rightarrow y_{train}$. The network learns to reject the noise in the shape-index map and produces masks and boundaries of the cell like structures in the shape-index map. The trained network was tested over 87 cases of labelled bright-field images, that were previously unseen by the network leading to a segmentation accuracy of 0.76 computed with **Jaccard, 1912** coefficient.

## Preprocessing

Preprocessing the input image to enhance the edge contrast and homogenising intensities helps in obtaining a good segmentation via MiSiC. Some of the preprocessing that gave good results are gamma correction for homogenising, unsharp masking for sharpening the image and sometimes a gaussian of laplace of the image that removes intensity variations in the entire image and keeps edge-like features.

## Parameters: scale and noise variance

The dataset one used to train the MiSiC contains cells with a width in the range of 8–10 pixels.Thus, to obtain a satisfactory segmentation, the input image must be scaled so that the average bacteria width is around 10 pixels. However, the scaling often modifies the original image leading to a smoother shape index map. Since, MiSiC has basically learnt to distinguish between smooth curvatures with well-defined boundaries from noisy background. A smooth image without inherent noise leads to a lot of false positive segmentations. Therefore, counterintuitively, synthetic noise must be added to the scaled or original image for a proper segmentation. It must be kept in mind that the noise variance should not reduce the contrast of the edges in the original image while it should be enough to discard spurious detections. Gaussian noise of a constant variance may be added to the entire image or alternatively, the variance could be a function of the edges in the input image.

## *Myxococcus xanthus* and *Escherichia coli* classification

A U-NET was trained on dataset two to segment a single channel phase-contrast image into an image containing semantic classification of each species. The probability map for each label is color coded (blue = *E. coli. coli*, orange = *M. xanthus*) such that each pixel has a probability value to be part of a given class. In rare instances, bi-color objects are obtained because in these cases the prediction is not homogeneous inside the predicted objects. For subsequent analysis, these objects were filtered for

the morphometric analysis shown in *Figure 5c*. A trained model and a python script illustrating how it was used is available at: https://github.com/pswapnesh/MyxoColi.

## Image analysis and statistics

The validation of the automated MiSiC results shown in *Figures 2 and 3* and *Figure 2—figure supplement 1*, was performed by comparing MiSiC masks to hand-annotated masks obtained by a single observer, herein referred to as ground-truth. This ground truth was also compared to data annotated by a second observer to test variability between observers. For comparison, curves of Jaccard Index vs. IoU threshold were constructed based on *Jeckel and Drescher, 2021*. Such curves allow a precise estimate of the accuracy and quality of shape detection and prediction.

For this, the $IoU = \frac{X \cap Y}{X \cup Y}$ (X = Ground Truth object; Y = Predicted object ) was determined by overlaying each predicted MiSiC mask with its corresponding ground truth mask and the number of True Positives (TP), False Positives (FP) and False Negatives (FN) that remain above varying IoU thresholds, applied from values ranging from 0.00 to 1.00 in 0.01 intervals, was determined so as to calculate the Jaccard index defined by the ratio $JI = \frac{TP}{TP+FP+FN}$ for a given threshold value.

To estimate the accuracy of classification, JIs were determined directly by comparing hand-annotated (ground truth) datasets obtained from fluorescence image of GFP (*Myxococcus*) and mCherry (*E. coli*).

Analysis of MiSiC performance in the presence of noise and comparison with Supersegger.

To illustrate MiSiC performances in the presence of noise and in comparison with SuperSegger (*Stylianidou et al., 2016*), datasets consisting of 141 *E. coli* microcolony images were retrieved from the SuperSegger website. These images were analysed with the provided parameters with SuperSegger and with the following parameters with MiSiC: Cell width = 9, Scaling factor = Auto, Noise = 0.001, Unsharp = 0.6 and Gamma = 0.1. To assess the segmentation robustness to noise for each program, datasets were normalized by the maximum intensity value recorded in the first frame of the dataset and Gaussian noise was added with varying variance. The resulting datasets were then analysed with the initial parameters used to compute reference segmented images except for MiSiC where the noise parameter was set to 0. The relative performance of each program was then evaluated by computing Dice indices (*Zijdenbos et al., 1994*).

$$\text{Dice index} = \frac{2A \cap B}{A+B}$$

## Morphological analyses

Classic morphological features: Area, Perimeter, Bounding Box (Width, Height), Circularity, Feret diameter, minimum Feret diameter, MajorAxis (ellipse), MinorAxis (ellipse), (n) number of objects.

Special calculated morphological features: Solidity = Area / Convex Area; AR = MajorAxis / MinorAxis; Extend = Area / (Width*Height).

## Demograph construction

To construct the plot shown in *Figure 4e* (demograph), the cells bodies were obtained with MiSiC segmentation and the binary mask was analyzed with the MicrobeJ software (*Ducret et al., 2016*), with a cell model set to parameters (area >0.5 µm$^2$, Circularity <0.8, 'poles' = 2) to filter all remaining segmented objects that do not correspond to cells. The localization of the centroid (cell middle) and length of longitudinal axis was then determined for each cell under MicrobeJ. The fluorescent clusters were detected with a local maxima filter and their position relative to the middle of the cell was plotted along the axis with a positive sign. Negative sign clusters are therefore the manifestation of rare cells with bi-polar foci. The fluorescent clusters are plotted as dots with a color scale based on spatial density.

## Cell division detection

In *Figure 4i*, clusters of fluorescent protein FtsZ-NG were used as cell division markers. The clusters were detected by local maxima detection (scikits-image.peak_local_max(image = fluorescence image, label = MiSiC mask, num_peaks = 1)).

## Calculation of cell division ratios

The cell division ratio in *Figure 4j* was calculated using the spatial 2D density derived from (i) the mask of the total fluorescent cell population across the entire image and (ii), the mask of the total number of fluorescent maxima (reflecting dividing cells) across the entire image. Spatial densities were calculated with sklearn.neighbors.KernelDensity(), with a bandwidth of 1 % of the image width. The proportion of dividing cells was obtained by dividing the spatial density maps: density of fluorescent maxima/ density of total cells.

## Code availability

A MiSiC pip installable python package is available at https://github.com/pswapnesh/MiSiC (*Swapnesh, 2021*; copy archived at swh:1:rev:45f659124a8e207f78296d77664fb96de5472708).

A Graphic user Interface pip installable as python package is available at https://github.com/leec13/MiSiCgui (*Espinosa, 2021*; copy archived at swh:1:rev:97401bedd44aa7b28d22f8cf87e76b521f15f40a).

## Acknowledgements

We thank Anke Treuner-Lange in Lotte Sogaard-Andersen lab for sharing the plasmid pKA32 with us (*Treuner-Lange et al., 2013*). MiSiC robustness was tested on microscope images showing various bacterial strains that were kindly shared with us. For that, we thank Anne Galinier and Thierry Doan (*Bacillus subtilis*), Christophe Bordi (*Pseudomonas aeruginosa*), Aretha Fiebig (*Caulobacter crescentus*) and Romain Mercier (SgmX-GFP images of *Myxococcus xanthus*). We also thank Baptiste Piguet-Ruinet and Célia Jonas for constructing the pDM14 plasmid during their internship. We thank Jean Raphael Fantino for the design of the MiSiC Logo. SP, LE, and TM are funded by CNRS within a 80-Prime initiative. TM is funded by an ERC Advanced Grant (JAWS EM is funded by an AMIDEX-PhD program from Aix-Marseille University). MN, JBF, AL and SR are funded by two ANR grants IBM (ANR-14-CE09-0025-01) and HiResBacs (ANR-15-CE11-0023).

## Additional information

### Competing interests

Tâm Mignot: Reviewing editor, *eLife*. The other authors declare that no competing interests exist.

### Funding

| Funder | Grant reference number | Author |
| --- | --- | --- |
| ERC advanced grant | JAWS 885145 | Tâm Mignot |
| AMIDEX | | Eugénie Martineau |
| ANR | IBM (ANR-14-CE09-0025-01) | Marcelo Nöllmann |
| ANR | HiResBacs (ANR-15-CE11-0023) | Marcelo Nöllmann |
| CNRS 80-prime | | Swapnesh Panigrahi |

The funders had no role in study design, data collection and interpretation, or the decision to submit the work for publication.

### Author contributions

Swapnesh Panigrahi, Conceptualization, Investigation, Methodology, Software, Visualization, Writing – original draft, Writing – review and editing; Dorothée Murat, Investigation, Validation; Antoine Le Gall, Validation, Visualization; Eugénie Martineau, Data curation, Methodology; Kelly Goldlust, Conceptualization, Data curation, Investigation, Methodology, Validation, Visualization, Writing – original draft, Writing – review and editing; Jean-Bernard Fiche, Marcelo Nöllmann, Leon Espinosa, Conceptualization, Data curation, Funding acquisition, Investigation, Methodology, Project administration, Supervision, Validation, Visualization, Writing – original draft, Writing – review and editing;

Sara Rombouts, Tâm Mignot, Conceptualization, Data curation, Funding acquisition, Investigation, Methodology, Project administration, Supervision, Validation, Writing – original draft, Writing – review and editing

## Author ORCIDs
Dorothée Murat (iD) http://orcid.org/0000-0001-5809-9267
Leon Espinosa (iD) http://orcid.org/0000-0002-1923-2069
Tâm Mignot (iD) http://orcid.org/0000-0003-4338-9063

## Decision letter and Author response
Decision letter https://doi.org/10.7554/eLife.65151.sa1
Author response https://doi.org/10.7554/eLife.65151.sa2

---

## Additional files

### Supplementary files
- Transparent reporting form
- Supplementary file 1. Table S1 - Bacterial strains used in this study.
- Supplementary file 2. Table S2 - Primers.
- Supplementary file 3. Table S3 - Plasmids.

### Data availability
The tensorflow model describe in this article is available in GitHub: https://github.com/pswapnesh/MiSiC (copy archived at https://archive.softwareheritage.org/swh:1:rev:45f659124a8e207f78296d77664fb96de5472708); https://github.com/leec13/MiSiCgui (copy archived at https://archive.software-heritage.org/swh:1:rev:97401bedd44aa7b28d22f8cf87e76b521f15f40a). Source data files have been provided for Figures 2, 3, 4 and 5.

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
