## [Decision Letter]

**Acceptance summary:**

Your work of developing the MiSiC package to segment single cells in crowded bacterial colonies and identify different species in the colonies is of great importance to the community. The final version of the manuscript incorporated all reviewers' comments with improved readability. An updated user guide is provided. We are also pleased with your commitment to disseminate this important tool to other research labs in the community.

**Decision letter after peer review:**

Thank you for submitting your article "MiSiC, a general deep learning-based method for the high-throughput cell segmentation of complex bacterial communities" for consideration by *eLife*. Your article has been reviewed by 3 peer reviewers, one of whom is a member of our Board of Reviewing Editors, and the evaluation has been overseen by Gisela Storz as the Senior Editor. The following individual involved in review of your submission has agreed to reveal their identity: Zach Hensel (Reviewer #2).

The reviewers have discussed their reviews with one another, and the Reviewing Editor has drafted the following to help you prepare a revised submission.

Essential revisions:

All three reviewers recognize the significance and potential impact of the work and would like to see this work being implemented and disseminated to a wider bacterial cell biology field. However, the following essential revisions are required before the work could be considered for publication at *eLife*. These essential revisions were discussed and agreed upon among the reviewers and the reviewing editor during the consultation. In addition to these essential revisions, detailed comments from the three reviewers should also be addressed in the revision.

1. Please benchmark the performance of MiSiC against currently available segmentation methods using similar approaches such as Supersegger, DeepCell and DeLTA.

2. Please provide a detailed description of the working principle of MiSiC, and a ready-to-use workflow (or a handbook/guide) for users. These descriptions should include not only the essential steps, but also detailed parameters lists/ranges/considerations, such as cell density, size, pixel size, signal-to-noise ratio, and cell shape range etc. The goal of this essential revision is to ensure that the MiSiC tool can be easily disseminated and implemented by other non-technical-driven users in order to maximize the impact of the work.

*Reviewer #1 (Recommendations for the authors):*

1. It is surprising to see that the authors have not applied the same algorithm to oval or round-shaped cells. Based on the principle of SIM, I do not see why MiSiC cannot be applied to those cells. Could the authors comment further on the limitations or show some results of round cells?

*Reviewer #2 (Recommendations for the authors):*

The results applying U-Net to the SIM images are very interesting and I look forward to seeing if it will improve performance of the segmentation method (based on DeLTA) that we are using in our lab now; alone or input together with the unprocessed image.

*Reviewer #3 (Recommendations for the authors):*

– The authors should emphasize that MiSiC is a 2D image analysis tool. It cannot handle 3D image data and it cannot handle 3D+time image data natively at the moment. This needs to be clearly stated.

– The introduction is rather short and does not provide sufficient context for most readers in my opinion. Particularly the 2nd paragraph is so short that the arguments are not clear. The second sentence criticizes machine learning techniques for requiring large amounts of training data. But the method presented in this paper is also a machine learning technique that requires a large amount of training data. I recommend that the authors significantly expand this paragraph to clarify and motivate their methodology.

– More generally, the introduction would benefit by placing the paper into the context of other image analysis tools for bacterial segmentation and colonies. There has been a lot of activity in image analysis for microbiology recently, and I think it would be helpful to readers to learn about this context.

– Results, paragraph 1 +2: Can the authors explain why the images need to be scaled so that the average cell size is set to 10 pixels? I guess this is based on the implicit assumption that there are sufficient intensity gradients on the 2x2 pixel scale used by the Hessian. As the SIM approach is a critical component of the MiSiC method, a clear explanation is needed here.

– A major advantage of MiSiC is that it uses the SIM images as training data for the CNN, which seems to result in a trained CNN model that can produce accurate segmentations for phase contrast, brightfield and fluorescence images. Is this true? Can the authors please very clearly state if this is true? The statements regarding this point in the manuscript are not completely unambiguous in my opinion. If true, this would be a major advance for bacterial image segmentation. Therefore, the authors should also state if the trained CNN model results in equal performance on all 3 imaging modalities. Or does a different CNN model need to be trained for each of the 3 imaging modalities with the same MiSiC workflow?

– Figure 2: In order for MiSiC to perform better on non-rod-shaped bacteria (filamentous bacteria, spirochetes, or cocci) – would the user need to generate new training data and re-train the model? I think this needs to be clarified.

– The authors supplemented their manually annotated data with synthetic data created by images of model rods with synthetic noise. Can the authors explain why this was done? Is the training with the manual annotation not sufficient? If the authors only used this synthetic data, does MiSiC also produce accurate segmentations, or is the real data needed?

– The semantic segmentation obtained by MiSiC (Figure 4) is impressive and works well. It is unclear whether this semantic segmentation also works in cases of strong intermixing between the cell types. Can the authors comment on that?

– Any development of a single cell segmentation method should include a graph of the Jaccard coefficient (and/or Dice index) as a function of the intersection over union, with error bars. The authors need to add such a graph to the manuscript so that authors can judge the quality of the segmentation.

[Editors' note: further revisions were suggested prior to acceptance, as described below.]

Thank you for submitting your article "MiSiC, a general deep learning-based method for the high-throughput cell segmentation of complex bacterial communities" for consideration by *eLife*. Your article has been reviewed by 3 peer reviewers, including Jie Xiao as the Reviewing Editor and Reviewer #1, and the evaluation has been overseen by Gisela Storz as the Senior Editor. The following individual involved in review of your submission has agreed to reveal their identity: Zach Hensel (Reviewer #2).

The reviewers have discussed their reviews with one another, and wished the authors to provide the following before the work can be accepted for publication:

1. Update User Guide to address issues and miscellaneous bugs related to file installations, encoding, handling of multiple images and etc as specified in reviewers 1 and 2's comments.

2. Expand the introductory or Discussion section to include a more detailed comparison with existing algorithms with a focus on the strength and weakness of each method, as specified in Reviewers 1 and 3's comments.

*Reviewer #1 (Recommendations for the authors):*

The revised manuscript by Panigrahi et al. addresses our major concerns. We are especially pleased to see the handbook created for the github page. Not only will the handbook be a valuable resource to users, but also the handbook can be updated as improvements and additions are made to the MiSiC GUI. Also, the new supplemental figure S1 is a helpful illustration of the two user-set parameters. We understand the difficulty in comparing MiSiC to different methods and appreciate the quantification of MiSiC's performance. Outlining the appropriate uses and limitations of MiSiC in the discussion is also appreciated. We hope the authors could further address the following concerns. The goal is to maximize the usability of MiSic for the community.

1. Cell density should be explicitly addressed in 1a of the handbook to match the heading. The new example of Caulobacter in Figure 3a is less dense than the images from the previous manuscript version, indicating that MiSiC can segment less dense cell populations. Since the majority of the images are still very dense clusters, it is still worth addressing density in the handbook if not in the manuscript text.

2. Suggesting specific preprocessing methods is useful, but the names alone might not be enough detail for the target MiSiC user. Either referencing FIJI plugins that accomplish the recommended processing or adding citations to the methods section would clarify these suggestions.

3. MiSiC is not expected to be absolutely accurate, and sometimes the binary mask output will require manual edits. For example, two cells recognized as one might need to be manually separated by pixels. If the GUI could incorporate the ability to manually modify the mask, that would tremendously increase the functionality of MiSiC. However, if that is too labor intensive, the post-processing section of the handbook should address how to make these manual adjustments to the output. This could also be mentioned in the discussion.

4. Could the authors expand the comparison between MiSiC with DeltaT, DeeptCell and Supersegger in the third paragraph of introduction be incorporating some of the language in the rebuttal letter? The goal is to give a bird's eye view of the current field so readers will have a clear assessment of which is good for what and understand MiSiC's uniqueness better.

5. There are some errors in GUI. They are not related to the segmentation algorithm, but confusing for users sometimes. For instance, the cell width measuring function is not always returning the right measurement. The way to break this function is to draw an extremely long line before actually tracing the short axis of any cell. Furthermore, MiSiC should be able to analyze all images belong to the same directory with just "one click", in theory. However, this was not the case today when we fed the GUI with multiple images. New masks could not be generated unless previous/existing images and masks are all cleared from the workspace. We also realized that applying the same parameter settings based on only one image does not guarantee accurate segmentation for other images, even though these images belong to the same experiment/run. It would be great if the authors could fix these bugs to enhance user experience.

*Reviewer #2 (Recommendations for the authors):*

I think that the authors have sufficiently addressed issues raised in the reviews. The examples of preprocessing/parameter choices in the handbook will be very useful.

In my opinion, the availability workflows to generate synthetic data and train the models would strengthen the manuscript since some potential users will want to sacrifice general for specific performance. However, anyone with the expertise to do that also has the expertise to reinvent the wheel to some degree based on what is reported in the manuscript.

The only other issue I have now is installation instructions (MiSiCgui page) no longer work for me (Windows 10; following instructions for conda). I created a python environment as specified, installed the misic package, and tried to install the GUI. (1) I think that the "use package" instructions should be updated, because add_noise for example is now in extras.py; (2) The GUI pip install command raised an error regarding file encoding. I don't know whether the 2nd error is specific to my system and did not spend much time trying to diagnose it. Lastly, the screenshots on the github page are the old version (noise=0.0001 rather than 1).

*Reviewer #3 (Recommendations for the authors):*

Author's response to the general comment 1 (page 1+2 of the rebuttal letter): I now understand better how it can be difficult to benchmark MiSiC against the other segmentation software. I also appreciate that the authors now discuss these other tools in lines 73-84 of the manuscript. However, the essential points that make the other tools unsuitable for the analysis that was desired by the authors are not mentioned in the main text (only in the rebuttal letter). For me, as a potential user of all of these tools, and probably for anyone who reads such a paper, it is important to know the strengths and weaknesses of these tools and why the other tools are not suitable for the authors' application. Therefore, I recommend that the authors should expand further the relevant paragraph in the main text, to more clearly describe why the other tools are not suitable. This doesn't have to be overly critical of the other tools, but it would be helpful to the readers.

All other comments were addressed nicely by the authors in my opinion.

---

## [Author Response]

Essential revisions:All three reviewers recognize the significance and potential impact of the work and would like to see this work being implemented and disseminated to a wider bacterial cell biology field. However, the following essential revisions are required before the work could be considered for publication at eLife. These essential revisions were discussed and agreed upon among the reviewers and the reviewing editor during the consultation. In addition to these essential revisions, detailed comments from the three reviewers should also be addressed in the revision.1. Please benchmark the performance of MiSiC against currently available segmentation methods using similar approaches such as Supersegger, DeepCell and DeLTA.

These software packages are machine (or Deep) learning-based but they do not address similar needs and none was designed or proven to be able to perform semantic segmentation on cells with different shapes. DeLTA was designed for the tracking of *E. coli* cells over many generations constrained in a so-called microfluidics device (called the mother machine) where cells are aligned on a microfluidics channel. Thus, DeLTa was not shown to work for semantic segmentation and is likely too specialized for a broad application.

Supersegger is an alternative to MiSiC for the segmentation of bacteria in dense colonies. Before developing MiSiC, we actually evaluated Supersegger extensively to attempt segmentation of *Myxococcus* cells in dense environments. During this experience, we observed that to produce acceptable results with Supersegger we had to train the network for each individual field-of-view (FOV). This was a lengthy process that required manual segmentations to produce reliable ground truths for each FOV and therefore made Supersegger unusable for the segmentation of large experiments with tens of FOVs displaying highly heterogeneous cell densities, as this would have required the separate training of each FOV for each experiment. We have never attempted to segment both Myxococcus and *E. coli* cells with Supersegger as this software tool was not designed to handle several cell types with different shapes at once.

DeepCell is a general-purpose tool developed mainly for the segmentation of eukaryotic cells, not of bacteria. Nevertheless, we installed DeepCell and attempted to segment a typical field of view containing Myxococcus and *E. coli* cells using the most closely applicable network provided by DeepCell (i.e. NuclearSegmentation). Unfortunately, the network was unable to converge to a prediction. The most likely reason is that DeepCell was developed to detect a single kind of eukaryotic cell at a time, not to perform semantic segmentation on cells with different shapes. We did not test DelTa as its design was specific to detect cells in microfluidic channels.

Our conclusion from this limited testing is that it is unlikely that software packages that were designed for very different purposes, and that were not designed or shown to work for semantic segmentation, would perform efficiently. Of course, it is likely that intense network retraining and/or parameter tuning would improve performance of these algorithms, but even in that case this would only demonstrate that these methods are ill suited for the semantic segmentation of cells in dense environments at high throughputs.

A major asset of MiSiC is its simple use, efficiency under several image modalities, ability to identify multiple bacterial species and robustness to noise and intensity variations for complex dataset segmentation. For these reasons, we think that will be a preferable alternative to the aforementioned softwares for these applications. To make this clear and rather than conducting disputable analyses between softwares for this revision, we rigorously quantified MiSiC performance by comparing it to the ground truth using an established statistical method as recommended by the reviewers. These analyses now demonstrate that MiSiC performs well under various modalities and on numerous species. In addition, it also allows comparing performance for the various conditions and species and thus an estimate of its advantages and limitations. This allows the user to evaluate the use of MiSiC for a given application, a type of analysis which to our knowledge has not been provided by the other softwares. The manuscript is now deeply modified in many instances to discuss the points raised in this response.

2. Please provide a detailed description of the working principle of MiSiC, and a ready-to-use workflow (or a handbook/guide) for users. These descriptions should include not only the essential steps, but also detailed parameters lists/ranges/considerations, such as cell density, size, pixel size, signal-to-noise ratio, and cell shape range etc. The goal of this essential revision is to ensure that the MiSiC tool can be easily disseminated and implemented by other non-technical-driven users in order to maximize the impact of the work.

Thank you for this recommendation. We now provide a detailed handbook for the correct segmentation of bacterial cells under MiSiC, including advice for installation under Napari, pre-processing, segmentation and post-segmentation analyses.

Reviewer #1 (Recommendations for the authors):1. It is surprising to see that the authors have not applied the same algorithm to oval or round-shaped cells. Based on the principle of SIM, I do not see why MiSiC cannot be applied to those cells. Could the authors comment further on the limitations or show some results of round cells?

We now provide data showing that the quality of the segmentation decays as cell shape deviates from the rod shape (see Figure 3). This is clear for *Bacillus* filaments, which are well resolved but only partially septated and for *Caulobacter*, which is crescent shaped and segmented by MiSiC with an accuracy inferior to that of typical rod-shaped like *E. coli*, *Myxococcus* or *Pseudomonas*. As for round cells, the initial training data was such that round objects (see Author response image 1) would be excluded since it is frequent in images that there are spurious background objects that are also round. Thus, training the network to detect round cells is certainly feasible and will indeed make it useful to segment round bacteria, but it will also increase the spurious detections of background objects. For these reasons, we have not implemented round cells’ detection in this version.

**Author response image 1. sa2fig1:** Note the presence of a round object (perhaps a round cell) in the PC image. MiSiC excludes this object because it deviates too much from the training data set only to retain the rod shape objects.

Reviewer #3 (Recommendations for the authors):– The authors should emphasize that MiSiC is a 2D image analysis tool. It cannot handle 3D image data and it cannot handle 3D+time image data natively at the moment. This needs to be clearly stated.

This is correct, we now make this unambiguously clear in the introduction and refer to other approaches for 3D analyses.

– The introduction is rather short and does not provide sufficient context for most readers in my opinion. Particularly the 2nd paragraph is so short that the arguments are not clear. The second sentence criticizes machine learning techniques for requiring large amounts of training data. But the method presented in this paper is also a machine learning technique that requires a large amount of training data. I recommend that the authors significantly expand this paragraph to clarify and motivate their methodology.

Agreed. We significantly expanded this part to contextualize the study better in terms of existing approaches and needs that motivated the development of MiSiC;

– More generally, the introduction would benefit by placing the paper into the context of other image analysis tools for bacterial segmentation and colonies. There has been a lot of activity in image analysis for microbiology recently, and I think it would be helpful to readers to learn about this context.

Agreed. We now discuss recent image analysis tools that allow bacterial cell segmentation in 2D but also 3D communities with added references to Hartmann, 2021 and Zhang et al. 2020.

– Results, paragraph 1 +2: Can the authors explain why the images need to be scaled so that the average cell size is set to 10 pixels? I guess this is based on the implicit assumption that there are sufficient intensity gradients on the 2x2 pixel scale used by the Hessian. As the SIM approach is a critical component of the MiSiC method, a clear explanation is needed here.

This is now explained in the text. Briefly, the cell size must be set to pixels to approximately match the width of the artificial cells used to enrich the training data set.

– A major advantage of MiSiC is that it uses the SIM images as training data for the CNN, which seems to result in a trained CNN model that can produce accurate segmentations for phase contrast, brightfield and fluorescence images. Is this true? Can the authors please very clearly state if this is true? The statements regarding this point in the manuscript are not completely unambiguous in my opinion. If true, this would be a major advance for bacterial image segmentation. Therefore, the authors should also state if the trained CNN model results in equal performance on all 3 imaging modalities. Or does a different CNN model need to be trained for each of the 3 imaging modalities with the same MiSiC workflow?

This comment was also raised by reviewer 1. We now provide quantification of how “true” this is. The results now provided in a new Figure 2b show that a single CNN resolves bacterial cells in CP, Fluo and BF with efficiency comparable to the naked eye and almost similar accuracy. Although clearly images are more accurately segmented in the following order: Fluo, CP and BF.

– Figure 2: In order for MiSiC to perform better on non-rod-shaped bacteria (filamentous bacteria, spirochetes, or cocci) – would the user need to generate new training data and re-train the model? I think this needs to be clarified.

As shown in the new Figure 3b, with our general training method, the accuracy of segmentation declines as cell shape deviates from rod-shapes. Nevertheless the system still accurately segments curved cells such as *Caulobacter* cells. For filamentous cells, as shown in Figure 3—figure supplement 1, using post processing methods like watershed, it is possible to refine to cell masks that also detect cell separations within the filaments. As discussed in our response to reviewer 1, round cell detection will require fine tuning of the network with a small number of new annotated data. In this version we omitted that data to avoid the detection of spurious non-cell objects. Given that training uses synthetic data, it is quite straightforward to implement new training data in MiSiC based on the needs, which we now explain in text.

– The authors supplemented their manually annotated data with synthetic data created by images of model rods with synthetic noise. Can the authors explain why this was done? Is the training with the manual annotation not sufficient? If the authors only used this synthetic data, does MiSiC also produce accurate segmentations, or is the real data needed?

Unet-based segmentation methods generally require annotated datasets that sample the diversity of images that may be encountered in ‘real-life’ for segmentation. In our case, synthetic data was mainly used for the following reasons:

a) Generalization of segmentation model to unseen imaging modalities and bacterial shapes. The ability to use synthetic data circumvents the necessity of a well annotated dataset that spans over different experimental conditions (different imaging modalities, bacterial shapes, background noise and spurious objects ). Such a dataset is necessary for an end-to-end deep learning model (raw image input and segmented image output). The synthetic data also adds examples of various cell densities that may be encountered in unseen images.

b) Smooth cell shapes: Secondly, the synthetic masks help the decoder side of U-Net to fit a smooth cell shape to the segmented bacteria.

The “real annotated” data is nevertheless necessary: when only trained with synthetic data, the segmentation works but it is not as accurate as with additional ‘real images’. This is presumably due to the fact that the synthetic data was generated by randomly throwing cell-shaped objects onto an image. In this case, the intricate patterns created by cell-cell interactions in ‘real-life’ situations are not sampled. Hence, ‘real’ images, even from a single imaging modality, are required to ‘capture’ the patterns created by dense cell communities. An alternative could have been to use a synthetic cell model with cell-cell interactions, but this is an active field of research and beyond the scope of this work.

We now explain the benefits of this approach in the methods section.

– The semantic segmentation obtained by MiSiC (Figure 4) is impressive and works well. It is unclear whether this semantic segmentation also works in cases of strong intermixing between the cell types. Can the authors comment on that?

We believe that by “strong intermixing” the reviewer refers to the areas where *Myxococcus* cells actively penetrate the *E. coli* colony and interact very tightly with the *E. coli* cells as shown for example in zoom inset in Figure 5c. Our analysis in a Figure 5b and Figure 5d shows that indeed classification works well for a large majority of the population. There are nevertheless ambiguous classifications which are observed as a minor overlap between the two populations. These ambiguities are also apparent in the zoomed image Figure 5c where it is visible that a few classifications events are uncertain. This is inevitable in the tight interaction areas due to low contrast in these areas. The ambiguous cells can be easily eliminated from analysis by filtering with discriminating parameters as shown in Figure 5d. We now discuss these limitations in the text.

– Any development of a single cell segmentation method should include a graph of the Jaccard coefficient (and/or Dice index) as a function of the intersection over union, with error bars. The authors need to add such a graph to the manuscript so that authors can judge the quality of the segmentation.

See our answer above for this particular point, which we believe is a major addition in this revision.

[Editors' note: further revisions were suggested prior to acceptance, as described below.]

Reviewer #1 (Recommendations for the authors):The revised manuscript by Panigrahi et al. addresses our major concerns. We are especially pleased to see the handbook created for the github page. Not only will the handbook be a valuable resource to users, but also the handbook can be updated as improvements and additions are made to the MiSiC GUI. Also, the new supplemental figure S1 is a helpful illustration of the two user-set parameters. We understand the difficulty in comparing MiSiC to different methods and appreciate the quantification of MiSiC's performance. Outlining the appropriate uses and limitations of MiSiC in the discussion is also appreciated. We hope the authors could further address the following concerns. The goal is to maximize the usability of MiSic for the community.1. Cell density should be explicitly addressed in 1a of the handbook to match the heading. The new example of Caulobacter in Figure 3a is less dense than the images from the previous manuscript version, indicating that MiSiC can segment less dense cell populations. Since the majority of the images are still very dense clusters, it is still worth addressing density in the handbook if not in the manuscript text.

Thank you for the comment. The handbook now addresses how noise adjustment can help segment images with low cell densities:

“MiSiC can segment bacterial cells at both high and low density. Importantly, in low cell density images, the smoothening of background noise after scaling (see main text ) can lead to spurious cell detection, which can be resolved by adding synthetic noise (see section 1d to set up the noise parameter).”

2. Suggesting specific preprocessing methods is useful, but the names alone might not be enough detail for the target MiSiC user. Either referencing FIJI plugins that accomplish the recommended processing or adding citations to the methods section would clarify these suggestions.

The handbook now links these methods to the FIJI software.

3. MiSiC is not expected to be absolutely accurate, and sometimes the binary mask output will require manual edits. For example, two cells recognized as one might need to be manually separated by pixels. If the GUI could incorporate the ability to manually modify the mask, that would tremendously increase the functionality of MiSiC. However, if that is too labor intensive, the post-processing section of the handbook should address how to make these manual adjustments to the output. This could also be mentioned in the discussion.

Incorporating these tools to the GUI could be interesting in a future version, but it is not immediately necessary as other existing softwares provide these means to correct the MiSiC masks.

We now mention these methods in more details in the handbook:

“The MiSiC mask can occasionally contain predicted cells with imperfect separation or poorly detected septa. […] Manual correction is also feasible in that software using a very practical GUI and associated toolbox that allows editing the contours directly on the mask.”

As well as in the Discussion section lines 443-449:

“Along similar lines, non-cell or improperly separated cell objects an appear in the MiSiC masks and while some can be removed by the introduction of noise, an easy way to do it is to apply a post-processing filter using morphometric parameters to remove objects that are not bacteria. […] MicrobeJ and Oufti are especially useful from downstream analyses of the masks because they both allow cell tracking as well as determining protein localization at sub-diffraction resolution.”

4. Could the authors expand the comparison between MiSiC with DeltaT, DeeptCell and Supersegger in the third paragraph of introduction be incorporating some of the language in the rebuttal letter? The goal is to give a bird's eye view of the current field so readers will have a clear assessment of which is good for what and understand MiSiC's uniqueness better.

We have now introduced these what these softwares provide and explained their merits as well as their current limitations justifying the development of MiSiC. This new text is now in the introduction Lines 72-101.

**“**Machine-learning based techniques are powerful alternatives to overcome the above limitations of traditional segmentation approaches. […] Inspired by these methods, we decided to develop MiSiC (Microbial Segmentation in dense Colonies), a general CNN-based tool to segment bacteria in single or multispecies 2D bacterial communities at very high throughput.”

5. There are some errors in GUI. They are not related to the segmentation algorithm, but confusing for users sometimes. For instance, the cell width measuring function is not always returning the right measurement. The way to break this function is to draw an extremely long line before actually tracing the short axis of any cell.

We have not been able to reproduce this bug, so we could not fix it.

Rare bugs like these may be environment-specific, we now wrote in the handbook that upon such problems users should feel free to contact us directly.

Furthermore, MiSiC should be able to analyze all images belong to the same directory with just "one click", in theory. However, this was not the case today when we fed the GUI with multiple images. New masks could not be generated unless previous/existing images and masks are all cleared from the workspace.

For multiple images, we chose to provide a « process all » option that allows the segmentation of an image stack rather that multiple image files. This is especially convenient to process images of time lapse recordings. For multiple independent images the users just need to assemble them as an stack, for example under ImageJ/FIJI. We added a “Processing of multiple images” paragraph in the handbook to make this clear.

We also realized that applying the same parameter settings based on only one image does not guarantee accurate segmentation for other images, even though these images belong to the same experiment/run. It would be great if the authors could fix these bugs to enhance user experience.

Unfortunately, even when images belong to the same experiment they are not identical and setting up the parameters on a single image does not guarantee that all images will be segmented with similar accuracy. Nevertheless, we found that for an image stack from a time lapse or from images of a same experiment it is often possible to set “mean_width” and “noise” parameters that will be largely effective for the entire stack. For this several parameter combinations might need to be tested to find the best compromise. If the images vary a lot across a single experiment it might be a solution to split them in distinct related image sets for better prediction.

We added tips for the segmentation of image stacks in the “processing of multiple images” paragraph of the handbook to make this clear.

Reviewer #2 (Recommendations for the authors):I think that the authors have sufficiently addressed issues raised in the reviews. The examples of preprocessing/parameter choices in the handbook will be very useful.In my opinion, the availability workflows to generate synthetic data and train the models would strengthen the manuscript since some potential users will want to sacrifice general for specific performance. However, anyone with the expertise to do that also has the expertise to reinvent the wheel to some degree based on what is reported in the manuscript.

We also think that anyone with the computational skills to train the CNN with a new data set will be able to generate this set without being directed to a specific workflow.

The only other issue I have now is installation instructions (MiSiCgui page) no longer work for me (Windows 10; following instructions for conda). I created a python environment as specified, installed the misic package, and tried to install the GUI.

To correct this problem, we have now removed all dependencies of MiSiCgui on the MiSiC package. However the installation of the MiSiC package in the same environment as MiSiCgui may raise an incompatibility. Thus, in some cases it will be necessary to remove the old environments of MiSiC and MiSiCgui and force pip install to re-install the package:

pip install --upgrade
--force-reinstall git+https://github.com/leec13/MiSiCgui.git

We have added this tip to the handbook and the GitHub page.

(1) I think that the "use package" instructions should be updated, because add_noise for example is now in extras.py;

Correct, the "use package » was updated to include the extras.py module in the imports that provide the add_noise function.

(2) The GUI pip install command raised an error regarding file encoding. I don't know whether the 2nd error is specific to my system and did not spend much time trying to diagnose it. Lastly, the screenshots on the github page are the old version (noise=0.0001 rather than 1).

Correct we fixed these issues.

Reviewer #3 (Recommendations for the authors):Author's response to the general comment 1 (page 1+2 of the rebuttal letter): I now understand better how it can be difficult to benchmark MiSiC against the other segmentation software. I also appreciate that the authors now discuss these other tools in lines 73-84 of the manuscript. However, the essential points that make the other tools unsuitable for the analysis that was desired by the authors are not mentioned in the main text (only in the rebuttal letter). For me, as a potential user of all of these tools, and probably for anyone who reads such a paper, it is important to know the strengths and weaknesses of these tools and why the other tools are not suitable for the authors' application. Therefore, I recommend that the authors should expand further the relevant paragraph in the main text, to more clearly describe why the other tools are not suitable. This doesn't have to be overly critical of the other tools, but it would be helpful to the readers.All other comments were addressed nicely by the authors in my opinion.

Thank you for the positive feedback and final recommendation. See our answer to reviewer 1 to see how we modified the text to introduce the available tools.